# Intravascular Complications of Central Venous Catheterization by Insertion Site in Acute Leukemia during Remission Induction Chemotherapy Phase: Lower Risk with Peripherally Inserted Catheters in a Single-Center Retrospective Study

**DOI:** 10.3390/cancers15072147

**Published:** 2023-04-04

**Authors:** Marco Picardi, Claudia Giordano, Roberta Della Pepa, Novella Pugliese, Maria Esposito, Davide Pio Abagnale, Maria Luisa Giannattasio, Dario Lisi, Martina Lamagna, Francesco Grimaldi, Giada Muccioli Casadei, Mauro Ciriello, Marcello Persico, Gianpaolo Gargiulo, Fabrizio Pane

**Affiliations:** 1Hematology Unit, Department of Clinical Medicine and Surgery, Federico II University Medical School, Via Sergio Pansini, 5, 80131 Naples, Italy; 2Department of General Surgery, Endocrinology, Orthopaedics, and Rehabilitations, Federico II University Medical School, Via Sergio Pansini, 5, 80131 Naples, Italy

**Keywords:** central venous catheters, PICC, acute leukemia, induction chemotherapy regimens

## Abstract

**Simple Summary:**

Central vein catheter (CVC) insertion is a main risk factor for deep vein thrombosis and blood stream infection in patients undergoing induction chemotherapy for acute leukemia. The decision of the treating physician to catheterize the basilica/brachial vein site as the frontline central vascular access has an important effect in minimizing morbidity and likely health care costs related to CVC complications in hematologic patients with severe and prolonged neutropenia.

**Abstract:**

The basilic/brachial (BBV), internal jugular (IJV), and subclavian veins (SCV) are commonly used as central venous catheter (CVC) sites. A BBV approach [peripherally inserted central catheter (PICC)] is increasingly used for short- to intermediate-term CVCs for acute leukemias undergoing cytotoxic intensive regimens. In this retrospective study, the catheterization of the BBV, IJV, and SCV in patients with previously untreated acute leukemia was assessed. The primary outcome was the composite incidence of catheter-related symptomatic deep-vein thrombosis (sDVT) and bloodstream infection (BSI) from catheterization up to 30 days later. In a 10-year period, 336 CVC were inserted in the BBV (*n* = 115), IJV (*n* = 111), and SCV (*n* = 110) in 336 patients suffering from AML (*n* = 201) and ALL (*n* = 135) and undergoing induction chemotherapy. The primary outcome events were 8, 20, and 27 in the BBV, SCV and IJV cohorts (2.6, 6.9, and 9.6 per 1000 catheter-days, respectively; *p* = 0.002). The primary outcome risk was significantly higher in the IJV-cohort than in the BBV-cohort (HR, 3.6; 95% CI, 1.6 to 7.9; *p* = 0.001) and in the SCV-cohort than in the BBV-cohort (HR, 2.6; 95% CI, 1.2 to 5.9; *p* = 0.02). PICC was a valid CVC for the induction chemotherapy of acute leukemia for the lowest risk of sDVT and BSI.

## 1. Introduction

In the upper arm basilic/brachial vein (BBV), internal jugular vein (IJV), and subclavian vein (SCV), central catheterization is associated with infectious, thrombotic, and mechanical/malfunction complications in patients with hematological malignancies [1,2]. Thanks to some specialized viscous matrix molecules, some bacteria and fungus adhere to the fibronectin and other serum components that coat catheters. Pathogens may enter the bloodstream when they exceed a crucial threshold in the biofilm, that the protein sheath creates, leading to catheter-related (CR) blood stream infection (BSI). This condition may be complicated by septic thrombophlebitis, massive pulmonary emboli, metastatic infections, and/or septic shock, especially in the neutropenic phase following cytotoxic agent intensive regimens for acute leukemia [3]. Deep vein thrombosis (DVT) and BSI are the major adverse events related to the central venous catheter (CVC) and should not be considered separate entities, given their bidirectional relationship [1,2,3]. Mechanical complications including arterial puncture, bleeding, neurological injury, and/or pneumothorax may also occur in association with catheter positioning [4]. Finally, dislocation, occlusion, and/or rupture could be a possible cause of CVC malfunction and early removal [1,2,3]. All these complications have a significant effect on morbidity, mortality, and health-care costs and should therefore be minimized [5]. However, only few data are available regarding patients characterized by frequent events of severe neutropenia [1,2,3]. In patients with acute myeloid leukemia (AML) and acute lymphoblastic leukemia (ALL), the anatomical decision of which vein to catheterize for the delivery of supportive medication has been contentious [1,2,6]. International clinical practice guidelines often leave it up to the treating physician to decide which vascular access to catheterize [7,8].

This retrospective study was conducted on adult patients who had been admitted to the Hematology Unit for acute leukemia and scheduled to receive antineoplastic treatment, with the aim of identifying the best frontline central venous access in this setting characterized by an expected duration of chemotherapy-induced severe aplasia, i.e., absolute neutrophil count (ANC) < 500/mm^3^ and platelets < 10,000/mm^3^, of ≥7 days [1,2,3]. Based on our previous prospective controlled trials on a similar group of patients [3,9,10], we hypothesized that the risk of CR-major complications during the entire phase of the hematological remission induction would differ according to the site of catheter insertion. We explored CR-adverse-event incidence during a fixed period in the three principal clinical settings of central vascular catheterization, i.e., BBV, IJV, and SCV, in critically ill patients requiring urgent therapy with intensive chemotherapy and maximal supportive care for untreated AML and ALL. The primary endpoint was the composite incidence of CR-symptomatic (s) DVT and BSI from catheterization to 30 days later. The secondary endpoints were the rates of mechanical and malfunction complications of CVC.

## 2. Patients and Methods

### 2.1. Study Design and Oversight

In the database registry of the Hematology Unit at Federico II University Medical School in Naples (Italy), all subjects with acute leukemia who were referred to our center for primary catheter positioning for intensive chemotherapy of hematological remission induction between 1 January 2012 and 31 December 2021 were almost sequentially enrolled. For the analysis, we considered patients with centrally inserted central catheters (CICCs), i.e., implanted in the cervical-thoracic area (IJVs and SCVs) and patients with peripherally inserted central catheters (PICCs), i.e., implanted in the upper arm area (BBVs). All subjects were judged suitable candidates for central venous vascular access, with an expected use of ≥30 days, by the physicians inserting the catheter. More than one catheter per patient could not be included in the assessment.

All necessary approvals were obtained from the Ethics Committee of the Federico II University Medical School of Naples (Italy). Given the retrospective design, the acquisition of all individually informed consents was waived (except for implanting the CVC).

### 2.2. Eligibility Criteria and Participants

In our institution, CVC insertion was considered a routine interventional practice for all patients scheduled to receive cytotoxic agent regimens for hematological malignancies [3,9,10]. Clinical judgment was used to choose the BBV, IJV, or SCV as the best venous access site for catheterization. The collected central venous access data regarded patients who (i) were 18 years of age or older, (ii) were admitted to the Hematology Unit for untreated AML and ALL (diagnosed according to the World Health Organization Classification) [11], (iii) had nontunneled CVC inserted for the first time in the BBV (upper arm), IJV, or SCV, (iv) underwent cytotoxic agent-intensive regimens for hematological remission induction with a duration of chemotherapy-induced severe aplasia of ≥7 days [3,9,10], and (v) had a follow-up with an in situ CVC placement of at least 30 days after catheterization (except for patients who died or had the catheter removed for complications before Day 30).

### 2.3. Implantation Procedures

Staff physicians who had performed at least 50 previous procedures formed the team that inserted the catheters in the Intensive Care Unit or in the Hematology Unit at the patients’ bedsides. Usually, the catheterization was performed 24 h before the start of chemotherapy. For all participants, the international guidelines for preventing CR-complications, as already described, were followed [12,13,14]. All CVCs were inserted with the strict application of dedicated protocols for the safe insertion of central venous access. Surgical hand antisepsis, sterile gloves, surgical long sleeve gowns, hats, and masks were employed as part of the strictest sterile barrier measures. Sterilized drops were applied to the patients. No catheters were infused with antibiotics or antiseptics. The Seldinger procedure or a modified Seldinger technique was used to catheterize patients with the use of ultrasound. All CVCs were implanted using a suture-free device, and the location of the tip in the lower third of the superior vena cava or the cavoatrial junction was determined using an intracavitary electrocardiographic approach. The responsible doctor routinely looked for evidence of local inflammation, infection, or thrombosis at the insertion site. All patients receiving induction chemotherapy for acute leukemia received an antibacterial prophylaxis with levofloxacin and an antifungal prophylaxis with either posaconazole or fluconazole, in accordance with institutional standards [15,16,17].

No drugs were used for thrombosis prophylaxis.

### 2.4. Collected Data

The patients’ primary features and relevant laboratory data were acquired and recorded in a special computerized database. Furthermore, comprehensive data on catheter placement was acquired, including the kind of CVC, the order of CVC insertion (first), the location and side of insertion, the tip position, and any potential CR complications. The infusion, type, and length of daily usage were noted throughout the follow-up to determine the rationale for utilizing CVC. Transfusions, complete parenteral nutrition, anti-infective or anti-thrombotic medications, and chemotherapy were all recorded, as were the prevalence, persistence, and severity of neutropenia (<500/mm^3^) and/or thrombocytopenia (platelet count < 10,000/mm^3^).

### 2.5. Outcomes

The primary endpoint was the cumulative incidence of major CR-complications from the placement of the catheter (typically 24 h before the initiation of chemotherapy) until 30 days later. The combination of CR-sDVT and BSI in each implanted group was considered a major complication.

According to the CTCAE (National Cancer Institute Common Terminology Criteria for Adverse Events, Version 4), the DVT was grade 3 or above. The deep veins of the upper arm or cervical-thoracic region (basilica, brachial, axillary, subclavian, internal jugular, and/or brachiocephalic veins) were involved, and DVT happened suddenly on the ipsilateral side of the device implantation. The diagnosis of symptomatic thrombotic complications was based on symptoms and objective clinical signs and was confirmed in all cases by ultrasonographic criteria according to the institutional guidelines [3,10]. Initial physical examination findings indicative of thrombosis included discomfort, induration, erythema, exudates, and/or asymmetric venous distension [18,19,20]. Next, the DVT component of the main outcome was the diagnosis verified by ultrasonography. The ultrasonographic results included non-compressibility, lack of respiratory fluctuation, and/or the presence of a distinct peri-catheter thrombus, which is an intravascular mass that is echogenic and measures more than 0.5 cm from the CVC to the vessel wall.

The CR-BSI episodes included in the study were those defined as grade 3 or higher, according to the CTCAE [1,7]. When an infection was clinically suspected, blood cultures were routinely taken from the CVC and concurrently from a peripheral site, especially in cases of neutropenic fever (defined as ANC < 1000/mm^3^ with at least one temperature measurement of ≥38.0 °C) [15,16,17]. The diagnosis of CR-BSI required the detection of the same pathogen, both in a blood culture and at the catheter tip, or a differential time to a positivity of ≥2 h in a pair of central and peripheral blood cultures (i.e., the growth of microbes from a blood sample derived from a central catheter hub ≥2 h before microbial growth was detected in a blood sample derived from a peripheral vein).

The secondary endpoints were the rates of major mechanical complications (grade ≥3 according to the modified CTCAE, with the modification that pneumothorax requiring chest-tube insertion was classified as grade 3 instead of grade 2) during CVC insertion, and catheter malfunction during follow-up.

The rates of catheter removals and deaths at Day 30 were also reported.

### 2.6. Statistical Analyses

Using the overall log-rank test, the incidence of the primary outcome was compared among the three-site insertion options of the scheme. The Kaplan-Meier technique was used to estimate the time-related dependent variable in the research, which was defined as the duration of time between the insertion of the CVC and the occurrence of CR-sDVT and BSI within a predetermined time frame, or until 30 days following catheterization. For patients who died or from whom catheters were removed for mechanical/malfunction complications, follow-up was censored at the day of death or CVC removal. The pairwise comparisons were conducted combining insertion-site groups in a two-choice scheme with the use of a Cox model. 

Risk variables for the incidence of CVC complications were found using Cox analysis. Using a variable admission requirement of *p* ≤ 0.10 and a variable retention criterion of *p* ≤ 0.05, stepwise Cox analysis was used to identify multivariable risk variables.

Results from Cox analysis are shown as hazard ratios (HR) with 95% confidence intervals (CI) and associated *p* values. The median and range for continuous variables were determined. The log-rank test, χ2 test, and *t*-test were used for the statistical analyses. Statistics were regarded as significant at *p* values < 0.05. Using R 3.6.0, statistical analysis was carried out.

## 3. Results

### 3.1. Participants and Recruitment

The data from 347 adult patients with untreated acute leukemia who had been CVC- implanted at the Hematology Department of the Federico II University of Naples (Italy) for cytotoxic agent-intensive regimen administration were screened from January 2012 to December 2021. Eleven patients did not meet enrollment criteria. In the first day from catheter positioning, seven patients accidently removed CVCs (5 PICCs, 2 CICCs) and two patients (1 with PICC, 1 with CICC) died from cerebral hemorrhage; in addition, two patients (1 with PICC, 1 with CICC) did not receive scheduled intent-to-cure therapy. During the screening procedure, all these cases were excluded for CVC non-adequate follow-up or non-intensive cytotoxic regimens. Finally, a total of 336 patients treated with intent-to-cure anticancer therapy, i.e., hematological remission induction chemotherapy, and a total of 336 in situ CVC placements with adequate follow-up were analyzed for the endpoints of the study. A flowchart of the present trial is shown in Figure 1.

Out of 336 patients, 115 (34%) had CVCs located in the BBV (BBV-cohort), 111 (33%) in the IJV (IJV-cohort) and 110 (33%) in the SCV (SCV-cohort). Table 1 shows patient characteristics at the baseline according to the site of catheter positioning. Overall, there were 170 males and 166 females with a median age of 53 years (range, 18–72 years). Age, sex, prothrombotic risk factors, underlying hematological malignant disease, eastern cooperative oncology group performance (ECOG) status and blood cell counts were well balanced among the three cohorts.

### 3.2. CVC Insertion and Use

The catheter and insertion procedure characteristics of the three cohorts are summarized in Table 2. More PICCs than CICCs were positioned in the Hematology Unit at the patients’ bedsides. All central intravascular accesses were inserted with the use of ultrasonography. The majority of cutaneous antiseptics utilized to disinfect the catheter insertion site were alcohol-based preparations. In the CICC placement setting (IJV + SCV: 221/336, 66%), the most frequent catheter positioned at IJV and SCV sites, respectively, was a 7 Fr device with a triple lumen (Arrow^®^, Teleflex, 3015 Carrington Mill Boulevard Morrisville, NC, USA). In the PICC placement setting, central intravascular access was obtained through the basilic vein in 89% of cases and the brachial vein in the remaining 11% of cases; the most frequent catheter positioned was a 5 Fr device with a double lumen.

The initial insertion attempt success rates were comparable across the three groups, demonstrating the team’s overall proficiency in inserting catheters into the three venous locations. In all patients, the catheter tips were positioned between the cavoatrial junction and the lower third of the superior vena cava. The IJV-implanted cohort underwent catheterization more promptly than any of the other two cohorts. All patients received chemotherapy infusions through the CVC, and the three cohorts received a variety of induction chemotherapy regimens that were evenly distributed.

For each of the three catheter insertion locations, the average use time was 30 days.

### 3.3. Post-Chemotherapy Aplasia

The median percentage of the received dose intensity of the cytotoxic agents was 94% in the BBV-cohort, 93% in the IJV-cohort, and 93% in the SCV-cohort. Chemotherapy-induced severe neutropenia (500/mm^3^) occurred in 100% of patients in the BBV-cohort, 100% of patients in the IJV-cohort, and 100% of patients in the SCV-cohort. The median duration of severe neutropenia was 20 days in each cohort with ranges of 8–22 days in the BBV-cohort, 8–23 days in the IJV-cohort, and 8–24 days in the SCV-cohort. Severe thrombocytopenia (<10,000/mm^3^) occurred in 100% of patients in the BBV-cohort, 100% of patients in the IJV-cohort, and 100% of patients in the SCV-cohort. The median duration of severe thrombocytopenia was 10 days in each cohort with ranges of 7–15 days in the BBV-cohort, 8–16 days in the IJV-cohort, and 7–14 days in the BBV-cohort.

### 3.4. Catheter-Related Symptomatic Deep-Vein Thrombosis and Blood Stream Infection

There were 55 primary outcome nonduplicate events (i.e., events that did not occur in the same catheter) in the three-site scheme comparison, and their incidence varied depending on the site of catheter insertion, with 8 events occurring in the BBV-cohort, 20 events occurring in the SCV-cohort, and 27 events occurring in the IJV-cohort (2.6, 6.9, and 9.6 per 1000 catheter-days, respectively; *p* = 0.0029); in particular, two patients from the BBV-cohort experienced both components of the composite endpoint (CR-sDVT and CR-BSI) almost simultaneously, but the event appearing first was considered as the primary outcome event (sDVT for both patients) for this analysis while they have been considered as separate for the evaluation of the single endpoint (Figure 2). In the pairwise comparisons in groups from the two-site schemes combined (Table 3), the risk of the primary outcome was significantly higher in the IJV-cohort than in the BBV-cohort (HR, 3.6; 95% CI, 1.6 to 7.9; *p* = 0.0016) and higher in the SCV-cohort than in the BBV-cohort (HR, 2.6; 95% CI, 1.2 to 5.9; *p* = 0.02), whereas the risk in the IJV-cohort was similar to that in the SCV-cohort (HR, 0.7; 95% CI, 0.4 to 1.3; *p* = 0.23).

CR-sDVT occurred in 6 patients in the BBV-cohort, 12 patients in the SCV-cohort and 15 patients in the IJV-cohort (5.2% vs. 11% vs. 13.5%, respectively). The incidence rate of sDVT was 1.9, 2.7 and 5.1 per 1000 catheter-days in the BBV-cohort, SCV-cohort, and IJV-cohort, respectively (*p* = 0.12). When thrombotic episodes occurred, the median interval between catheter insertion and thrombotic episodes was 16 days (range, 8–30 days) for the IJV-cohort and 20 days (range, 15–29 days) for the SCV-cohort versus 9.5 days (range, 7–10 days) for the BBV-cohort. The characteristics of the thrombotic episodes are presented in Table 4.

CR-BSI occurred in 4 patients in the BBV-cohort, 8 patients in the SCV-cohort and 12 patients in the IJV-cohort (3.5% vs. 7.2% vs. 10.8%). The incidence rate of BSI was 1.3, 4 and 4 per 1000 catheter-days in the BBV-cohort, SCV-cohort and IJV-cohort, respectively (*p* = 0.11). The median time between catheter implantation and BSI events, when they occurred, was 7 days (range, 6–8 days) for the BBV cohort, compared to 16 days (range, 8–24 days) for the SCV cohort and 15 days (range, 7–22 days) for the IJV cohort. Table 4 displays the type of pathogens that cause the disease. The pathogens that were most often isolated were coagulase-negative staphylococci.

### 3.5. Secondary Endpoints

The frequency of mechanical complications associated with catheter positioning in the three-choice comparison differed according to the insertion-site cohort (*p* = 0.035), with 6 events (bleeding, 3; neurologic damage, 3) in the BBV-cohort, 15 events (bleeding, 7; arterial puncture, 3; neurologic damage, 3; pneumothorax, 2) in the SCV-cohort, and 17 events (bleeding, 9; arterial puncture, 3; neurologic damage, 3; pneumothorax, 2) in the IJV-cohort. In the pairwise comparisons (Table 3), there were significantly fewer mechanical complications in the BBV-cohort than in the IJV-cohort (HR, 3.7; 95% CI, 1.5 to 9.5; *p* = 0.006), or SCV-cohort (HR, 3.1; 95% CI, 1.2 to 8.1; *p* = 0.017) but there was no significant difference between the IJV-cohort and SCV-cohort in pairwise comparison (*p* = 0.59).

Ten patients (8.6%) in the BBV-cohort (4 dislocations, 4 occlusions, and 2 ruptures), 11 patients (10%) in the IJV-cohort (4 dislocations, 4 occlusions, and 3 ruptures), and 12 patients (11%) in the SCV-cohort (5 dislocations, 5 occlusions, and 2 ruptures) had catheter malfunction (*p* = 0.85).

### 3.6. Thirty-Day Catheter Removals and Deaths

Fifteen patients (13%) in the BBV-cohort required catheter removal (median: 10 days after insertion). The reasons for PICC removals were malfunction in 10 patients, neurologic damage in three patients, and BSI complicated by septic thrombophlebitis in two patients (due to methicillin-resistant *Staphylococcus haemoliticus* in one and *Escherichia coli* in one, respectively). Nineteen patients (17%) in the IJV-cohort removed (median: 18 days after insertion) the device (malfunction in 8 patients, BSI complicated by septic thrombophlebitis in 8 patients [*Candida non-albicans* in 1, methicillin-resistant *Staphylococcus haemolyticus* in 3, methicillin-resistant *Staphylococcus epidermidis* in 2, *Staphylococcus aureus* in 1, and *Escherichia coli* in 1], and neurologic damage in three patients). Nineteen patients (17%) in the SCV-cohort removed (median: 20 days after insertion) the device (BSI complicated by septic thrombophlebitis in 8 patients [*Candida non-albicans* in 3, methicillin-resistant *Staphylococcus haemolyticus* in 2, methicillin-resistant *Staphylococcus epidermidis* in 1, *Staphylococcus aureus* in 1, and *Klebsiella pneumoniae* in 1], malfunction in 7 patients, and neurologic damage in 3 patients).

The 30-day mortality rate was 12% in the BBV-cohort (14 of 115 patients), 13% in the IJV-cohort (14 of 111 patients), and 14% in the SCV-cohort (15 of 110 patients). In the basilic/brachial group, death was from leukemia progression in 8 patients, persistent hypoplasia in three patients, pulmonary aspergillosis in two patients, and septic shock in one patient. In the internal jugular group, death was from leukemia progression in 8 patients, persistent hypoplasia in three patients, and septic shock in three patients. In the subclavian group, death was from leukemia progression in 9 patients, pulmonary aspergillosis in 3 patients, and septic shock in 3 patients.

### 3.7. Risk Factors for the Occurrence of CVC Complications

To identify risk factors for the occurrence of the primary endpoint, the impact of relevant variables at the time of CVC insertion (age, gender, prothrombotic risk factors, central catheter insertion site, type of acute leukemia, induction chemotherapy regimen, eastern cooperative oncology group performance status, neutrophils and platelets count), and post-chemotherapy severe and prolonged neutropenia were assessed in univariate and multivariate analyses, respectively. None of the preplanned subgroup studies revealed a significant interaction for the main outcome by univariate or multivariate analysis.

## 4. Discussion

The patient had to be in the greatest possible performance status in this critical situation in order to receive the planned anti-leukemic therapy, which includes induction and consolidation therapy and finally hematopoietic stem cell transplantation [21,22,23,24]. The adverse events related to the primary catheter positioning may negatively affect the acute leukemia cure, leading to the disruption of a dose-density/dose-intensity program. Thus, all efforts should be directed to minimize this risk.

Our study showed that in real-life CVC-related adverse events in acute leukemia patients undergoing intensive chemotherapy regimens for hematological remission induction remain a problem. Overall, the frequency of clinically relevant (thrombotic, infectious and mechanical) complications of grade 3 or higher reached a rate of 28%, i.e., about one out of three patients (Figure 3). While DVT and BSI may take longer to be identified and treated than bleeding and pneumothorax events, which in our study accounted for the majority of the differences in mechanical problems among insertion sites, the latter may be managed more quickly. The planned length of catheterization is crucial because, unlike mechanical issues, the cumulative risk of thrombotic and infectious complications rises with increased catheter exposure. The primary implanted CVC typically has an intended usage of at least 30 days; therefore the composite endpoint of the trial, i.e., sDVT and BSI incidence, was chosen primarily to evaluate the safety of the CVC.

To the best of our knowledge, this is the first large study in patients with hematologic malignancies experiencing neutropenia of grade 4 (according to the CTCAE) to identify the front-line insertion site of CVC as a risk factor for DVT and BSI. In a recent randomized controlled trial on CVC complications in cancer patients, Moss and colleagues demonstrated for patients receiving systemic anticancer therapy that totally implanted ports in comparison to PICCs were associated with a lower risk for CR-major adverse events [25]. However, in this trial, the total number of evaluated patients with acute leukemia was 39. Studies comparing significant CVC-related consequences in acute leukemia patients who had severe and persistent neutropenia after cytotoxic agent-intensive regimens are still rare. Until recently, patients with AML receiving induction chemotherapy for hematological remission were the subjects of the first randomized study comparing the two distinct insertion sites for frontline CVCs, CICC versus PICC [10]. However, this study was not designed to demonstrate a difference in DVT and BSI among the three different central venous access sites, i.e., BBV, IJV and SCV, mainly due to a limited sample size in each cohort.

The present study included an adequate number of acute leukemia cases and an adequate sample size of CVCs. It was characterized by 336 patients with AML (62%) and ALL (38%), with 336 in situ primary CVC (34% at BBV, 33% at IJV, and 33% at SCV) placements for front-line intent-to-cure anticancer therapy, with a median use of 30 days. Significantly, the catheterization of the basilic/brachial vein of upper arm was associated with a reduced risk of the combined outcome. CR-sDVT and BSI, the primary endpoint, had developed in 7% of patients in the BBV-cohort, 18% of patients in the SCV-cohort, and 24% of patients in the IJV-cohort, mostly with a reduced risk of at least 50 percentage points in the group implanted in the basilic/brachial-vein. This advantage of the basilic/brachial-vein catheterization in comparison with the jugular-vein and subclavian-vein catheterizations could be proven for all major adverse event categories investigated, that is, the frequency of sDVT and BSI, rate of sDVT and BSI per 1000 CVC days and per 1000 neutropenic CVC days (3, 9, and 12, in the BBV-cohort, SCV-cohort, and IJV-cohort, respectively), respectively.

DVT and BSI in the internal jugular group and subclavian group occurred later than for those in the basilic/brachial group, as clearly proved (Figure 2). These findings most likely reflect a number of reasons. One well-known key risk factor for CR-complications is neutropenia [1,2,3]. Patients with acute leukemia receiving induction chemotherapy typically have lengthy periods of neutropenia, and it has been observed that neutropenia lasting more than 20 days during hospitalization is a significant risk factor for both CR-BSI and DVT [1,2,3]. It may be inferred that both infectious and noninfectious triggers are involved in the pathogenesis of adverse outcomes at the CVC insertion site because in our series, the depth and duration of neutropenia were similar across the three implanted groups. Dix and colleagues found a substantially greater prevalence of DVT and BSI in patients with CVC implanted in IJV in a heterogeneous sample of patients with hematologic malignancies, which is consistent with our results [26]. The authors noted the development of facial hair and neck movement as causes of an ongoing dressing disturbance and long-term emergence of events. Before accessing the vein, the subclavian catheter often travels a long subcutaneous journey; in addition, the subclavian insertion site is relatively protected against dressing disruption. However, the subclavian insertion site has the greatest bacterial bioburden and a higher local temperature caused by the coverage through clothing [27]. According to reports, the skin of the middle part of the upper arm will have 50 to 100 colony-forming units/10 cm^2^, but the thoracic area would have 1000 to 10,000 colony-forming units/10 cm^2^ [10,27]. Furthermore, a variety of germs, particularly gram-positive bacteria, will have colonized the skin of the thoracic region [10,27]. As a further support for these features, Luft et al. described the role of exuberant skin colonization due to bacterial growth over time in the development of long-term CR-adverse events at the subclavian area [28].

Retrospective cohort studies [29,30] and metanalysis [31,32] have shown that patients with myeloid acute leukemia have a higher risk in suffering from upper extremity venous thrombosis (especially the brachial and basilic vein) and/or from BSI from gram positive bacteria [29,30,31,32]. First, PICC insertion is based on easier technical methods, with a minor trauma at the implantation site compared with CICC. Subclavian catheter insertion is more intrusive due to the mechanical stress that is inextricably linked to it. The production of inflammatory mediators and/or damage to the endothelium of the vein wall both contribute to thrombosis and increase the prothrombotic environment. Second, there is a marginal risk of luminal contamination since the cutaneous area in the middle third of the upper arm (PICC implantation site) has quantitatively fewer germs than the cervical-thoracic cutaneous area (CICC implantation site) (as reported above). Finally, choosing PICC means a lower upfront cost due to the reduced rate of infection and thrombosis, with, consequently, a reduced cost for the required medication/hospital admission days [10]. More research is needed to determine the burden and risk of problems associated with PICCs in patients with acute leukemia receiving high-dose chemotherapy. Consequently, well-designed research on the clinical applicability, safety, and benefits of employing PICC over CICC with a sufficient sample size might address this crucial issue, and we feel that this is the case with the current study.

This study suffers from some limitations: first, the retrospective nature which implies the lack of randomization. However, all these CVCs were inserted, cared for, and monitored in accordance with the same documented standard operating protocols, similar to a prospective research strategy [12,13,14]. The lack of information on the responsible physician selection process at a particular place of access may also pose difficulties. The differences in CR-DVT and BSI may be skewed by the choice of insertion location based on the clinical state or the severity of the patient’s disease, and it should be investigated if the patient’s clinical condition may have had an impact on this decision. Furthermore, some secondary endpoints of the study occurred at a similar frequency among the three implanted cohorts. However, the rate of mechanical complications at catheter positioning time was significantly lower in the PICC setting than in the CICC setting, especially owing to no events of arterial puncture and pneumothorax in the first. Finally, one could object to the decision of focusing our study on symptomatic thrombotic events but, in our real-life practice and also in several retrospective reports, sDVT and BSI are the major adverse events related to a central venous catheter that could influence hematological treatment and consequently, the outcome of the patient regarding morbidity, mortality, and also health-care costs [1,18,19,20]. Discussion of the clinical significance of asymptomatic CR-thrombosis is still ongoing. Regular ultrasonography screening has not often been advised to detect early thrombosis, and heparin treatment in the case of asymptomatic episodes remains debatable (for example, in presence of thrombocytopenia) [18,19,20].

## 5. Conclusions

In conclusion, our investigation revealed the initial CVC insertion in the IJV or SCV as the key risk factor for CR-DVT, and BSI in patients receiving induction chemotherapy for acute leukemia. Fortunately, no difference in catheter-related deaths was observed among the three cohorts of patients in this study. Thus, these results have not the strength to clearly recommend the basilica/brachial vein as the preferred insertion site for short- to-intermediate-term central venous catheterization. However, in general, the decision of the treating physician to catheterize the basilica/brachial vein site as the frontline central vascular access had an important effect in minimizing morbidity and likely health care costs related to CVC complications in hematologic patients with severe and prolonged neutropenia.

## Figures and Tables

**Figure 1 cancers-15-02147-f001:**
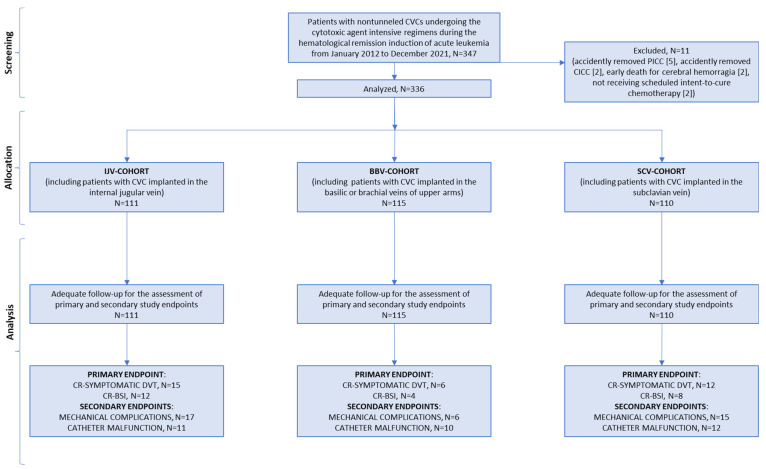
Diagram of the study. CVC, central venous catheter; PICC, peripherally inserted central catheter; CICC, centrally inserted central catheter; IJV, internal jugular vein; BBV, basilic/brachial vein; SCV, subclavian vein; CR, catheter-related; BSI, bloodstream infection; DVT, deep-vein thrombosis.

**Figure 2 cancers-15-02147-f002:**
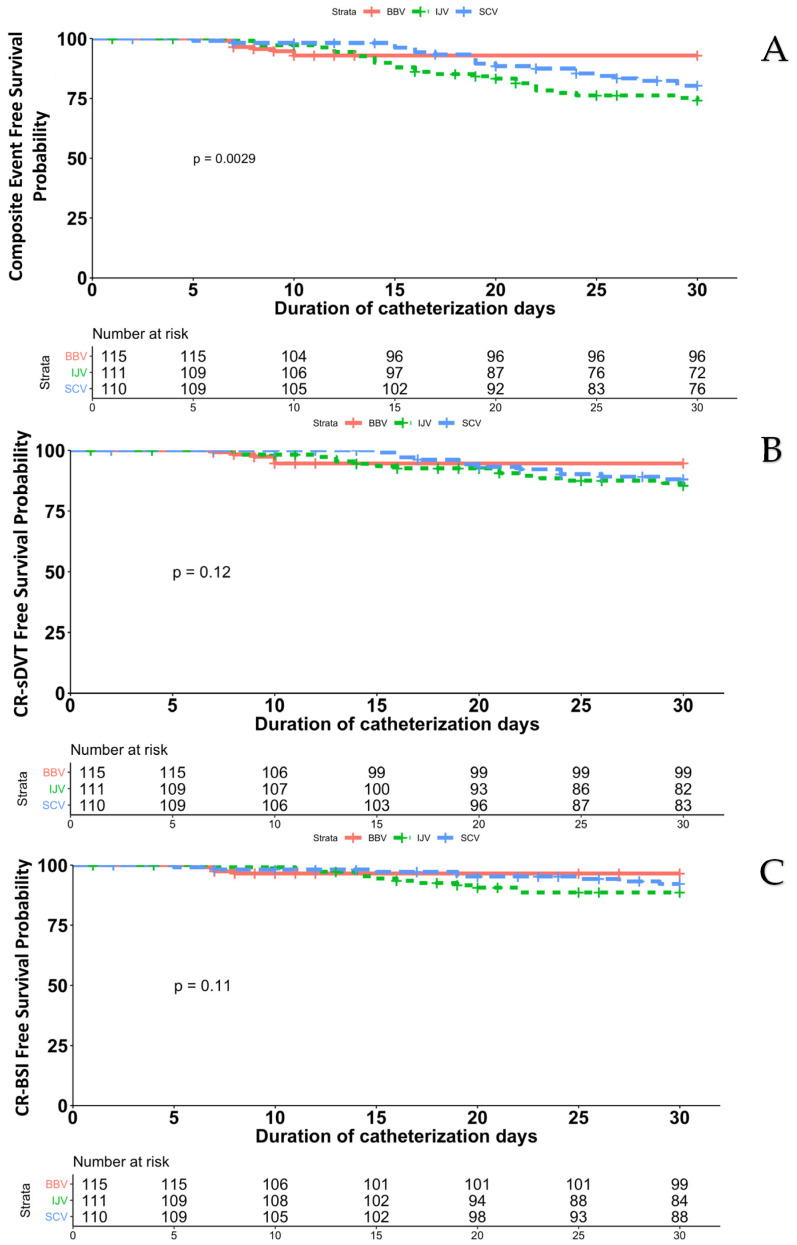
Rate of combined primary outcome events at 30 days after catheterization (**A**). Rate of catheter-related symptomatic deep-vein thrombosis (sDVT) at 30 days after catheterization (**B**). Rate of catheter-related bloodstream infection (BSI) at 30 days after catheterization (**C**). BBV, basilic/brachial vein; IJV, internal jugular vein; SCV, subclavian vein. Two patients from the BBV-cohort experienced both components of the composite endpoint (CR-sDVT and CR-BSI) almost simultaneously, but the event appearing first was considered as the primary outcome event (sDVT for both patients) for this analysis while they had been considered as separate for the evaluation of the single endpoint.

**Figure 3 cancers-15-02147-f003:**
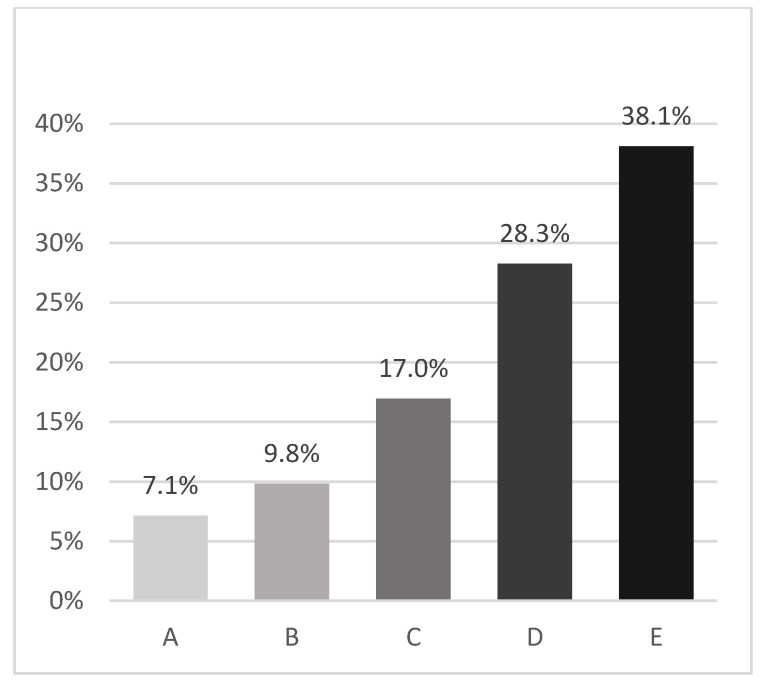
Composite incidence (%) of catheter-related clinically relevant complications. BSI, Bloodstream infection. DVT, Deep venous thrombosis. A—Catheter-related BSI; B—Catheter-related DVT; C—A + B; D—Mechanical complications + C; E—Catheter malfunction + D.

**Table 1 cancers-15-02147-t001:** Patients’ characteristics at baseline, according to the site of the catheter.

Variable	Total (%)	BBV-Cohort (%)	IJV-Cohort (%)	SCV-Cohort (%)	*p* Value
No. of patients	336	115	111	110	
Median age, years Range	53 18–72	53 18–72	54 18–72	54.5 22–70	0.71
Male	170 (51)	58 (50)	56 (51)	56 (51)	0.67
Prothrombotic risk factors					
BMI > 25	36 (11)	13 (11)	12 (11)	11 (10)	0.96
Smoke	60 (18)	22 (19)	20 (18)	18 (16)	0.85
Hypertension	69 (19)	23 (20)	24 (21)	22 (20)	0.97
Diabetes	26 (7)	9 (8)	10 (9)	7 (6)	0.71
Hematological disease					
AML	208 (62)	71 (62)	69 (62)	68 (62)	
ALL	128 (38)	44 (38)	42 (38)	42 (38)	0.98
ECOG status					
0–1	286 (85)	97 (84)	95 (85)	94 (85)	
2–3	50 (15)	18 (16)	16 (15)	16 (15)	0.97
Blood cell count					
WBC, ×10^3^/mm^3^					
Median, range	3.4, 0.2–147	3.2, 0.97–96	3.9, 1.0–147	3.5, 0.2–100	0.75
Neutrophils, ×10^3^/mm^3^					
Median, range	0.5, 0.4–12.34	0.45, 0.4–7.5	0.56, 0.8–12.34	0.48, 0.75–5.5	0.69
Hemoglobin, g/dL					
Median, range	9.5, 5.9–12.6	9.7, 5.9–12.1	9.5, 6.6–12.6	9.3, 6–12	0.81
Platelets, ×10^3^/mm^3^					
Median, range	41, 3.0–275	36, 3.0–275	42.5, 9.0–232	40, 5.0–200	0.92

Unless otherwise indicated, data refer to the number of patients. BBV: basilic and brachial veins; IJV: internal jugular vein; SCV: subclavian vein; BMI: body mass index; AML: acute myeloid leukemia; ALL: acute lymphoblastic leukemia; ECOG: eastern cooperative oncology group performance status; WBC: white blood cell.

**Table 2 cancers-15-02147-t002:** Characteristics of implanted catheters and their use.

Variable	Total (%) *n* = 336	BBV-Cohort (%) *n* = 115	IJV-Cohort (%) *n* = 111	SCV-Cohort (%) *n* = 110
Device insertion place				
Hematology ward bedside	120 (35)	92 (80)	17 (15)	11 (10)
ICU	216 (65)	23 (20)	94 (95)	99 (90)
Device type				
Single lumen	20 (6)	20 (17)	NA	NA
Double lumen	111 (33)	90 (78)	10 (9)	11 (10)
Triple lumen	205 (61)	5 (5)	101 (91)	99 (90)
4 French	40 (10)	40 (35)	NA	NA
5 French	70 (21)	70 (61)	NA	NA
6 French	5 (1)	5 (4)	NA	NA
7 French	177 (53)	NA	96 (86)	94 (85)
8 French	32 (9)	NA	15 (14)	16 (15)
Venous access				
Basilic	73 (22)	73 (63)	NA	NA
Brachial	42 (12)	42 (37)	NA	NA
Internal jugular	111 (33)	NA	111 (100)	NA
Subclavian	110 (33)	NA	NA	110 (100)
Right side	227 (67)	88 (76)	74 (67)	65 (59)
Left side	109 (33)	27 (24)	37 (33)	45 (41)
Attempts at venipuncture, n				
Median, range	1, 1–4	1, 1–4	1, 1–3	1, 1–3
Tip location				
Lower third of superior vena cava	84 (25)	22 (19)	31 (28)	31 (28)
Cavoatrial junction	252 (75)	93 (81)	80 (72)	79 (72)
Interval from CVC implantation to chemotherapy start, day				
Median	1	1	1	1
Range	0.5–1.5	0.5–1	0.5–1.5	0.5–1
Chemotherapy regimen				
Cytarabine-based	188 (56)	63 (55)	62 (56)	63 (57)
Fludarabine-based	20 (6)	7 (6)	7 (6)	6 (5.5)
MTX-asparaginase-based	128 (38)	45 (39)	42 (38)	41 (37)
Chemotherapy-induced hematological toxicity				
Severe neutropenia	336 (100)	115 (100)	111 (100)	110 (100)
Severe thrombocytopenia	336 (100)	115 (100)	111 (100)	110 (100)
Duration of catheterization—days				
Median	30	30	30	30
Range	1–30	6–30	1–30	2–30

Unless otherwise indicated, data refer to the number of patients. BBV: basilic and brachial veins; IJV: internal jugular vein; SCV: subclavian vein; ICU: intensive care unit. Cytarabine-based regimen included 3 + 7 protocol (cytarabine 200 mg/m^2^ in continuous *i.v.* infusion for 7 days and daunorubicin 60 mg/m^2^
*i.v.* on days 1, 3 and 5) ([21]). The fludarabine-based regimen included FLAG-Ida protocol (cytarabine 2000 mg/m^2^ daily as a 3-h *i.v.* infusion for 4 days plus fludarabine 30 mg/m^2^ daily as a 30-min *i.v.* infusion for 4 days plus idarubicin 12 mg/m^2^ daily as a 1-h *i.v.* infusion on days 2–4) ([22]). MTX-asparaginase-based regimen included the following: GIMEMA LAL-1913 (idarubicin, vincristine, dexamethasone, asparaginase, cyclophosphamide, cytarabine, 6-mercaptoturine, and methotrexate; for the specific schedules see [23]), and NILG ALL 10/07 (idarubicin, vincristine, dexamethasone, L-asparaginase, cyclophosphamide, cytarabine, 6-mercaptoturine, and methotrexate; for the specific schedules see [24]).

**Table 3 cancers-15-02147-t003:** Pairwise comparisons for the Trial Outcomes.

Outcome	Internal Jugular versus Basilic/Brachial (BB)				Subclavian versus Basilic/Brachial (BB)				Subclavian versus Internal Jugular			
	Jugular	BB	Hazard Ratio (95% CI) *	*p* Value	Subclavian	BB	Hazard Ratio (95% CI) *	*p* Value	Jugular	Subclavian	Hazard Ratio (95% CI) *	*p* Value
	Number				Number				Number			
Catheters	111	115			110	115			110	111		
Catheter-days	2812	3050			2902	3050			2812	2902		
Primary composite outcome **	27	8	3.6 (1.6–7.9)	0.0016	20	8	2.6 (1.2–5.9)	0.02	27	20	0.7 (0.4–1.3)	0.23
Symptomatic deep-vein thrombosis	15	6	2.6 (1.1–6.7)	0.032	12	6	2.1 (0.8–5.5)	0.15	15	12	0.8 (0.4–1.7)	0.53
Bloodstream infection	12	4	3.1 (1.0–9.6)	0.047	8	4	2.1 (0.6–7.0)	0.22	12	8	0.6 (0.3–1.6)	0.35
Secondary outcome												
Major mechanical complications	17	6	3.7 (1.5–9.5)	0.006	15	6	3.1 (1.2–8.1)	0.017	17	15	0.8 (0.4–1.7)	0.59
Arterial injury	3	-	-	0.99	3	-	-	0.99	3	3	0.9 (0.2–4.7)	0.95
Haematoma	9	3	3.9 (1.1–14.7)	0.04	7	3	2.9 (0.7–11.2)	0.12	9	7	0.7 (0.3–1.9)	0.52
Pneumothorax	2	-	-	0.99	2	-	-	0.99	2	2	0.9 (0.1–6.7)	0.95
Neurologic damage	3	3	1.1 (0.2–5.6)	0.88	3	3	1.2 (0.2–5.8)	0.85	3	3	0.9 (0.2–4.8)	0.96
Catheter malfunction	11	10	1.2 (0.53–2.9)	0.62	12	10	1.4 (0.6–3.2)	0.46	11	12	1.0 (0.5–2.4)	0.95

* Values in this column are hazard ratios unless otherwise indicated. NA denotes not applicable. Catheter malfunction included dislocations, occlusions, and ruptures. ** Two patients from the BBV-cohort experienced both components of the composite endpoint (CR-sDVT and CR-BSI) almost simultaneously but the event appearing first was considered as the primary outcome event (sDVT for both patients) for this analysis while they had been considered as separate for the evaluation of the single endpoint (Figure 2).

**Table 4 cancers-15-02147-t004:** The characteristics of catheter-related symptomatic deep vein thromboses and blood stream infections.

	Total (*n* = 336 Patients)	BBV-Cohort (*n* = 115 Patients)	IJV-Cohort (*n* = 111)	SCV-Cohort (*n* = 110)
Catheter-related deep vein thromboses				
Number of events	33	6	15	12
Thrombosis symptoms/clinical signs				
Yes	33	6	15	12
Ultrasonography diagnoses				
Yes	33	6	15	12
French thrombosed catheters				
5 Fr	6	6	-	-
7 Fr	12	-	7	5
8 Fr	15	-	8	7
Thrombus site				
Basilic vein	3	3	-	-
Brachial vein	3	3	-	-
Axillary vein	25	2	13	10
Subclavian vein	20	2	10	8
Internal jugular vein	7	-	5	2
Brachiocephalic vein	6	-	4	2
Thrombosis in multiple sites	25	4	11	10
Thrombus size (mm)				
Median (range)	20 (5–80)	20 (5–50)	25 (5–80)	20 (5–70)
Anti-thrombotic specific therapy *				
Yes	19	3	10	6
No	14	3	5	6
Catheter-related blood stream infections				
Number of events	24	4	8	12
Causative pathogens of blood stream infection				
Gram-positive	14	2	6	6
*Staphylococcus haemolyticus*	7	2	3	2
*Staphylococcus epidermidis*	4	-	2	2
*Staphylococcus aureus*	2	-	1	1
*Enterococcus* spp.	1	-	-	1
MDR gram-positive bacteria **	5	-	3	2
Gram-negative	6	2	1	3
*Escherichia coli*	3	2	1	-
*Klebsiella pneumonia*	3	-	-	3
MDR gram-negative bacteria ***	3	-	2	1
*Candida parapsilosis* ****	4	-	1	3
Antimicrobial prophylaxis				
Levofloxacin (500 mg daily orally)	336	115	111	110
Posaconazole (200 mg three times daily orally)	208	71	69	68
Fluconazole (200 mg daily)	128	44	42	42

Note: Unless otherwise indicated, data are the number of patients. BBV: basilic/brachial vein; IJV: internal jugular vein; SCV: subclavian vein. * Anti-thrombotic specific therapy, i.e., low-molecular-weight heparin, was introduced only for patients with a platelet count ≥ 20 × 10^3^/mm^3^. ** Among the 14 patients with gram-positive infection, there were 5 cases with multidrug-resistant (MDR) bacteria, i.e., methicillin-resistant *Staphylococcus* spp. (3 cases in the internal jugular group and 2 cases in the subclavian group). *** Among the 6 patients with gram-negative infection, there were 3 cases with MDR bacteria, i.e., extended spectrum β-lactamase (ESBL) producing *Escherichia coli* (*n* = 2, in the basilica/brachial group) and *Klebsiella pneumoniae*-carbapenemase (KPc)-producing (*n* = 1, in the subclavian group). **** Azole-resistant *Candida parapsilosis*.

## Data Availability

All data are available from the author upon request.

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
