# Peer review of "Intravascular Complications of Central Venous Catheterization by Insertion Site in Acute Leukemia during Remission Induction Chemotherapy Phase: Lower Risk with Peripherally Inserted Catheters in a Single-Center Retrospective Study"

_cancers, 2023, doi:10.3390/cancers15072147_

Round 1
Reviewer 1 Report
The authors present a retrospective study of complication rates of central venous catheters placed at different sites in acute leukemia patients. The study is of sufficient size to obtain meaningful data and is generally well written, with a very detailed study design and methodology. However, some of the figures need upgrading due to small fonts and the discussion requires some changes to better contextualize the data.
The manuscript needs some minor revisions:
1) Figure 1: Font is too small to easily read and needs reformatting
2) Figure 2: Figures and fonts are too small to read and see the main difference an d need to be revised.
3) Section 3.7 "Risk factors for the occurrence of CVC complications" only has one sentence and should be expanded to a short paragraph.
4) Figure 3: fonts are too small and needs to be revised for readability
5) The abbreviation CICC is used on page 14 and may not have been defined. Please check if this has been defined earlier in the text or add the full term in brackets here.
6) Page 15, 2nd full paragraph "PICC association with a great rate of major adverse events such as upper extremity venous thrombosis and/or BSI, especially in the case of acute leukemia, resulted from systematic reviews and/or meta analyses. Theoretically, these findings are startling because PICC employment might be safer than CICC's." This section is poorly worded and has scientific and grammar issues. Authors should consider summarizing the main relevant findings of the meta-analysis and systematic review, particularly where the current study deviates from these summary studies. Relevant references from the meta-analysis and/or systematic review should be selected for further comparison to the present study results.
7) The remainder of the 2nd paragraph which provides a rationale for why PICC line placement is/may be a superior route needs to be a separate paragraph and reworded. For example "Finally choosing a PICC means having a perceived lower upfront cost". Perception is not relevant, but rather state that it is lower cost if this can be justified in saying so. However, cost would be a minor consideration in choosing a route for a central line, so if this is kept in it needs to re-framed.
8) Page 16, first full paragraph, ""This study suffers of some limitations as the absence of a randomization procedure owing to the retrospective design." This is poorly worded and could be written: 'A limitation of this study was that it was retrospective rather than prospective and therefore lacked randomization.' After this the authors could expand on how retrospective study bias might lead to incorrect conclusions in this instance, but also discuss how the three study groups (BBV, IJV and SCV) had relatively equal sizes and similar characteristics as outlined in Table 1.
9) Page 16, first full paragraph: "It should be questioned whether the clinical condition of the patient might have influenced the selection of the insertion site; it cannot be excluded that differences in CR-DVT and BSI might be biased by the selection of the insertion site depending on clinical condition or the severity of the patient disease." This is a critical aspect of this study which the authors should actually try to address by trying to quantify the acuity of the patients and address if this is true or not. However, this would require further analysis.
10) Page 16, first full paragraph "Finally, the decision of focusing our study on symptomatic thrombotic events could be objectional. ...." This is a very weak way to end the paper. While this may be a valid issue, it should appear in the introduction and argued that despite this, thrombosis is a relevant end point which for this study.
Reviewer 2 Report
This is a single institution retrospective study evaluating central venous access in adult patients with acute leukemia receiving antineoplastic treatments. Methods and results clearly explained. Limitation of the study were pointed out. Discussion was well-organized and supported the study findings. I am interested to learn the incidence of asymptomatic DVTs. The authors reported the incidence of symptomatic DVTs in the patient population. I am also interested in learning how frequently patients had dressing changes as well as difference between dressing change frequency depending on whether patients had basilic/brachial vein versus internal jugular vein vs subclavian vein central catheterization.
Reviewer 3 Report
The availability of a central venous access is crucial for chemotherapy and supportive care in patients with acute leukemia.
In this paper the authors reported a single-center experience in the management of central venous catheters in a large cohort of leukemia patients, comparing the incidence of DVT, BSI and mechanical complications by site of insertion.
Despite some limitations, already underlined by authors, the study is interesting and seems to confirm the current practice to prefer PICC rather CICC.
I have just a couple of questions:
· How did they choose the site of insertions? Did the differences depend only by the year of diagnosis?
· Thrombosis seems to occur earlier in patients with BBV catheter than in those with IJV od SCV. How did the authors explain this fact?
· Besides basal prothrombotic risk factors there were other concomitant events favoring DVT, such as higher transfusion need, use of GCSF, rapid neutrophil recovery?
· Among CICCs, did they observe a relations with type of device?
